# An α2-Adrenergic Agonist, Brimonidine, Beneficially Affects the TGF-β2-Treated Cellular Properties in an In Vitro Culture Model

**DOI:** 10.3390/bioengineering9070310

**Published:** 2022-07-12

**Authors:** Megumi Watanabe, Tatsuya Sato, Yuri Tsugeno, Megumi Higashide, Masato Furuhashi, Araya Umetsu, Soma Suzuki, Yosuke Ida, Fumihito Hikage, Hiroshi Ohguro

**Affiliations:** 1Departments of Ophthalmology, School of Medicine, Sapporo Medical University, Sapporo 060-8556, Japan; watanabe@sapmed.ac.jp (M.W.); yuri.tsugeno@gmail.com (Y.T.); megumi.h@sapmed.ac.jp (M.H.); araya.umetsu@sapmed.ac.jp (A.U.); ophthalsoma@sapmed.ac.jp (S.S.); funky.sonic@gmail.com (Y.I.); fuhika@gmail.com (F.H.); 2Departments of Cardiovascular, Renal and Metabolic Medicine, Sapporo Medical University, Sapporo 060-8556, Japan; satatsu.bear@gmail.com (T.S.); furuhasi@sapmed.ac.jp (M.F.); 3Departments of Cellular Physiology and Signal Transduction, Sapporo Medical University, Sapporo 060-8556, Japan

**Keywords:** three-dimensional spheroid cell cultures, human trabecular meshwork (HTM), brimonidine, α2-adrenergic agonist, TGF-β2

## Abstract

We report herein on the effects of brimonidine (BRI), an α2-adrenergic agonist, on two-dimensional (2D) and three-dimensional (3D) cell-cultured TGF-β2-untreated and -treated human trabecular meshwork (HTM) cells. In the presence of TGF-β2 (5 ng/mL), (1) the effects of BRI on (1) the 2D HTM monolayers’ barrier function were investigated as estimated using trans-endothelial electrical resistance (TEER) measurement and FITC dextran permeability; (2) real-time analyses of cellular metabolism using a Seahorse Bioanalyzer; (3) the largeness and hardness of 3D spheroids; and (4) the expression of genes that encode extracellular matrix (ECM) proteins, including collagens (COL) 1, 4, and 6; fibronectin (FN) and α-smooth muscle actin (α-SMA); ECM modulators, including a tissue inhibitor of matrix proteinase (TIMP) 1–4; matrix metalloproteinase (MMP) 2, 9, and 14; and several endoplasmic reticulum (ER) stress-related genes, including the X-box-binding protein 1 (XBP1), the spliced XBP1 (sXBP1), glucose-regulated protein (GRP)78, GRP94, and CCAAT-enhancer-binding protein homologous protein (CHOP). BRI markedly inhibited the TGF-β2-induced increase in the values of TEER of the 2D cell monolayer and the hardness of the 3D spheroids, although it had no effect on their sizes. BRI also cancelled the TGF-β2-induced reduction in mitochondrial maximal respiration but had no effect on the glycolytic capacity. In addition, the gene expression of these molecules was quite different between the 2D and 3D cultures of HTM cells. The present observations found in this study indicate that BRI may beneficially affect TGF-β2-induced changes in both cultures, 2D and 3D, of HTM cells, although their structural and functional properties that were altered varied significantly between both cultures of HTM cells.

## 1. Introduction

Among the various anti-glaucoma medications, a selective α2-adrenoceptor agonist, brimonidine (BRI), induces a hypotensive effect, which results in a reduction in aqueous humor (AH) production by inhibiting the action of adenylate cyclase [1]. BRI also simultaneously stimulates the outflow of AH through the trabecular and uveoscleral pathways [1]. In addition to this, BRI also appears to have a neuroprotective effect [2,3,4,5]. While the mechanism responsible for this neuroprotective effect is not currently understood, it has been suggested that it involves the modulation of N-methyl-D-aspartate (NMDA) receptors. Such modulation would be expected to lead to apoptosis of the retinal ganglion cells (RGCs) and axonal degeneration by inhibiting excitotoxic effects by excitatory amino acids [6]. In fact, in rat models with experimentally increased intraocular pressure (IOP), the systemic administration of brimonidine was reported to prevent RGC cell loss but did not affect IOP [7]. These observations indicate that a selective α2-adrenoceptor agonist, such as BRI, would be expected to exert multiple potent effects, which would lead to a decrease in AH production, increased AH outflow, and neuroprotection. It is noteworthy that, among these, the BRI-induced effects on AH outflow are not well understood, especially in the case of the trabecular meshwork (TM), which is generally accepted as being the most important biological structure that regulates the facility of AH outflow [8]. It should also be noted here that previous studies reported that the α2-adrenoceptor is present in cultured human TM cells (HTM) [9] and that BRI affects the hyaluronidase activity in rabbit TM [10]. Therefore, based upon this collective body of evidence, it is of great interest to further investigate the BRI-induced effects on trabecular AH outflow using a reliable and proper in vitro model replicating the glaucomatous nature of HTM. Regarding such a study, we recently succeeded in establishing a three-dimensional (3D) HTM spheroid model with a multiple sheet structure by the presence of TGF-β2, a material that is well known to induce the formation of glaucomatous HTM cells [11].

Here, to study the effects of BRI in terms of glaucomatous TM, we performed the following measurements: (1) trans-endothelial electron resistance (TEER) and FITC dextran permeability of TGF-β2-treated two-dimensional (2D) cultured HTM cell monolayers; (2) cellular mitochondrial and glycolytic functions; (3) measurements of the physical nature of the TGF-β2-treated 3D spheroids, including the largeness and stiffness; and (4) the expression of major extracellular matrix (ECM) proteins: collagen (COL) 1, 4, and 6; fibronectin (FN) and α smooth muscle actin (αSMA); tissue inhibitors matrix proteinase (TIMP) 1–4; matrix metalloproteinase (MMP) 2, 9, and 14 (2D and 3D); and several genes that are related to endoplasmic reticulum (ER) stress.

## 2. Materials and Methods

### 2.1. Two-Dimensional and Three-Dimensional Cultures of Human Trabecular Meshwork (HTM) Cells

The immortalized cells of the HTM (Applied Biological Materials Inc., Richmond Canada) were maintained in a culture medium composed of HG-DMEM containing 10% FBS, 1% L-glutamine, and 1% antibiotic-antimycotic up to passage 20 according to the conventional 2D and 3D spheroid culture conditions for 6 days as followed in a previous report [11] after checking these cells were truly HTM cells [12]. All 2D and 3D cell cultures were repeated three times following the analyses. For the evaluation of the drug efficacy of brimonidine (BRI), an α2-adrenergic agonist on TGF-β2 (5 ng/mL)-treated HTM cells (2D and 3D) 10 μM BRI were supplemented on cell culture day 1. The drug concentrations were confirmed as the optimal concentrations based on previous studies [11,13,14], and a cell viability analysis of BRI using retinal pigment epithelium cells [15].

### 2.2. Barrier Function and Real-Time Cellular Metabolic Function Analyses of the 2D HTM Cells

The 2D HTM cells untreated or treated with 5 ng/mL TGF-β2 and/or 10 μM BRI were subjected to analyses of their barrier functions using TEER and FITC dextran permeability measurements as described previously [11,16]. Alternatively, the oxygen consumption rate (OCR) and the extracellular acidification rate (ECAR) of the 2D HTM cells under these conditions were measured as previously described using a XFe96 Seahorse Bioanalyzer (Agilent Technologies, Santa Clara, CA, USA) [17].

### 2.3. Measurements of the Largeness and Hardness of 3D HTM Spheroids

Based upon the configuration of the 3D HTM spheroid observed by phase contrast microscopy, the mean size of each 3D spheroid was determined as described previously [11,18]. Regarding the hardness of the 3D HTM spheroids, the required force (μN/μm) for compression of a single spheroid that induced 50% deformation over 20 s was measured as previously reported [11,18].

### 2.4. Immunocytochemistry of HTM Cells

The immunocytochemistry was performed as described in a recent report [19,20]. Briefly, the 2D and 3D cultured HTM cells as above were each fixed with 4% paraformaldehyde, permeabilized in 0.1% NP-40 in PBS for 10 min, and incubated successively with the first antibody: an anti-human rabbit antibody (1:200 dilutions) against COL1, 4, or 6 (ROCKLAND Antibodies & Assays, Limerick, PA, USA) or FN (Santa Cruz Biotechnology, Inc., Dallas, TX, USA) overnight, and a mixture of the second antibody: a goat anti-rabbit IgG (488 nm, 1:1000 dilutions, Invitrogen, Waltham, MA, USA), phalloidin (594 nm, 1:1000 dilutions, Cayman Chemical, Ann Arbor, MI, USA), and DAPI (1:1000 dilutions, DOJINDO, Osaka, Japan) for 3 h. Thereafter, the confocal immunofluorescent images were obtained.

### 2.5. Other Analyses

The real-time PCR using predesigned specific primers (Appendix A) and statistical analyses using Graph Pad Prism 8 (GraphPad Software, San Diego, CA, USA) were performed as described previously [11].

## 3. Results

To investigate the drug-induced effects of brimonidine (BRI) on glaucomatous TM, the TGF-β2-treated 2D- and 3D-cultured HTM cells, which replicate single sheet and multiple sheet constructions of the POAG human TM, respectively [21], were used. As shown in Figure 1, the barrier functions, as determined by the TEER and FITC dextran permeability measurements of the 2D HTM monolayers, indicated that TGF-β2 induced a substantial increase in the TEER levels and a relative decrease in the FITC dextran permeability, as was observed in previous studies [11,21], and these values were substantially reduced by the BRI treatment. To further evaluate the role of BRI on the biological cellular properties of TGF-β2-treated HTM cells, mitochondrial function and glycolytic system function measurements were carried out on 2D-cultured HTM cells using the Seahorse Bioanalyzer (Figure 2). The TGF-β2 treatment significantly decreased the maximal level of mitochondrial respiration and caused an increase in the glycolytic capacity in 2D-cultured HTM cells, as was observed in our recent study [17]. Interestingly, BRI cancelled the TGF-β2-induced reduction in the maximal mitochondrial respiration. In contrast, BRI had no effect on the TGF-β2-induced increase in the glycolytic capacity. These metabolic function analyses indicate that treatment with BRI prevents TGF-β2-induced mitochondrial dysfunction, potentially leading to beneficial changes in the cellular properties of TGF-β2-modulated 2D HTM cells.

Within the 3D HTM spheroid models, similar but also some different physical effects of BRI on TGF-β2-induced effects were also observed (Figure 3), which were thought to be a representative in vitro model that replicates multi-sheet layers of the HTM structure [11,21]. Specifically, the sizes of the 3D spheroids were reduced and they became harder as a result of the 5 ng/mL TGF-β2 or 10 μM BRI. In addition, BRI induced a significant reduction in the elevated TGF-β2-induced hardness of the 3D spheroids without synergist effects on the sizes of the 3D spheroids (Figure 3).

To study this further, qPCR and/or immunocytochemistry was performed to evaluate the expression of the ECM proteins; regulatory factors, including TIMP and MMP; and ER stress-related molecules. In the gene expression analysis, a TGF-β2-induced upregulation was observed for *COL1* (2D and 3D), *COL4* (2D), and *FN* (2D), BRI-induced up-regulation in the case of *COL1* (3D) and *COL4* (3D), and further enhancement of TGF-β2-treated *COL4* (3D) (Figure 4). Similar but much less effects by TGF-β2 and/or BRI were observed using the immunocytochemistry of the HTM cells (Figure 5; 2D and Figure 6; 3D). Similar to these differences between the 2D and 3D cultures, the gene expression of TIMPs, MMPs, and ER stress-related factors in the BRI were observed (Figure 7 and Figure 8). In 2D, TGF-β2 upregulated TIMP2 and 3, MMP2, and 14, and the TGF-β2 induced values for TIMP3 and 4, and CHOP were inhibited by BRI, and MMP9 was downregulated by BRI. In 3D, BRI downregulated TIMP3, MMP2 and MMP9, and CHOP in the presence or absence of TGF-β2. Thus, these observations suggest that the α2-adrenergic receptor agonist, BRI, has beneficial effects on POAG-related human TM by inhibiting the TGF-β2-induced increases in (1) the barrier functions in the 2D monolayers and (2) the hardness of the 3D spheroids, although these BRI-induced effects were measurably different between the 2D and 3D cultures in the HTM cells.

## 4. Discussion

A selective α2-adrenoreceptor agonist, BRI, has been identified to exert hypotensive effects that prevent IOP elevation after argon laser-assisted trabeculoplasty, in addition to the IOP control in patients with primary open-angle glaucoma (POAG) and ocular hypertension (OH) [22,23,24]. Possible mechanisms responsible for causing such dual effects include the possibility that BRI may induce a reduction in AH production and an increase in uveoscleral outflow [24]. In fact, it is well known that α2-adrenergic receptor agonists inhibit adenylyl cyclase activity via a Gi protein-dependent mechanism, resulting in a decrease in the cytosolic cAMP levels within the eye [9], and such signaling mechanisms have also been identified in the conventional 2D-cultured HTM cells [25]. Furthermore, we also identified that BRI induced significant inhibitory effects on the TGF-β2-related increases in the barrier function of 2D cells and the hardness of 3D spheroids. Similar to the BRI-induced inhibition of TGF-β2-related fibrosis in human Tenon’s fibroblasts in vitro [26], our results indicate that BRI causes TGF-β-stimulated ECM synthesis.

Functionally, our data indicate that BRI cancelled the TGF-β2-induced decrease in maximal mitochondrial respiration. It is known that TGF-β2 induces cellular metabolic reprograming and redox status, which is associated with cellular properties [27]. Indeed, we recently found that TGF-β2 shifts energy reserves from the mitochondria to glycolysis in both 2D and 3D HTM cells [17]. Taken together, our findings suggest that metabolic reprogramming is involved in the effect of BRI on TGF-β-induced ECM synthesis, and that changes in substrate utilization and oxygen consumption under different 2D and 3D culture conditions may have resulted in the formation of different BRI-induced cellular phenotypes.

At present, possible underlying mechanisms inducing such a significant suppression of the TGF-β2 effects on both 2D- and 3D-cultured HTM cells by BRI remain to be elucidated. However, these findings also suggest that BRI affects the expression of TIMPs and MMPs, based on a recent study reporting that BRI altered the expression of MMP9 and TIMP4 in human ciliary bodies, which express α2-adrenergic receptors [28]. Indeed, in the present investigation, we found that BRI caused substantial downregulation of TIMP3 and MMP2 and 9 in both the presence and absence of TGF-β2. In addition, BRI was found to induce a significant downregulation of CHOP and also caused relative decreases in other ER-stress-induced factors that were tested, although our knowledge regarding how BRI specifically affects ER stress remains limited. However, if BRI inhibits the TGF-β2-related effects on human TM, then it is indeed possible that it may also affect the TM-involved AH outflow facility and uveoscleral AH outflow, as has already been suggested [1].

As a study limitation of the current investigation, TGFβ2-treated HTM spheroids as an established in vitro model of glaucomatous TM represent quite reproducible experimental observations in terms of their physical properties, size, and stiffness as shown above. However, the characteristics of the HTM spheroids in the current study were different from our previous publication in terms of the gene expression of ECMs [11]. Specifically, there were no significant increases in the expression of COL1, 4, 6, and FN in the presence of TGFβ2, in contrast to the increases observed in our previous study [11]. Although we do not exactly know these differences, we speculated that some cell culture conditions such as FBS and other culture medium components were responsible, even though the same numbers of products and numbers of cell passages were used. However, even if there were such diversity between preparations, we believe that the current data regarding the BRI-induced effects was obtained because they were performed using the same preparations and repeated at least three times. Therefore, in conclusion, the currently observed new beneficial pharmacological effects of BRI on glaucomatous HTM may provide significant insight into the therapeutic strategy for antiglaucoma medication in patients with POAG. Furthermore, additional studies are required to obtain a comprehensive understanding of the molecular mechanisms of the BRI-induced effects on the dexamethasone-treated steroid-induced glaucoma TM model, in addition to the TGF-β2-treated POAG TM model, with certainty.

## Figures and Tables

**Figure 1 bioengineering-09-00310-f001:**
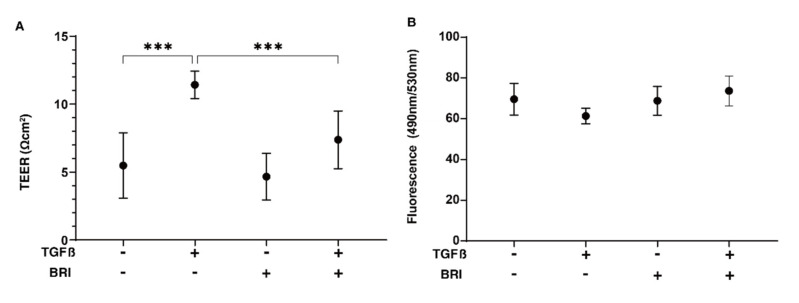
Effects of brimonidine (BRI) on the TEER and FITC dextran permeability of TGF-β2-treated 2D monolayers of HTM cells. The effects of brimonidine (BRI, 10 μM) on the barrier function (Ω cm^2^) of TGF-β2 (TGFβ, 5 ng/mL)-untreated (NT) or -treated 2D HTM cells were studied by TEER measurements (**A**) and FITC dextran permeability (**B**). “+” is reagents addition. “−” is reagents non-addition. Experiments were conducted in triplicate using freshly prepared samples (*n* = 4). Regarding the data, the mean ± standard error of the mean (SEM) are presented. *** *p* < 0.005; ANOVA followed by a Tukey’s multiple comparison test.

**Figure 2 bioengineering-09-00310-f002:**
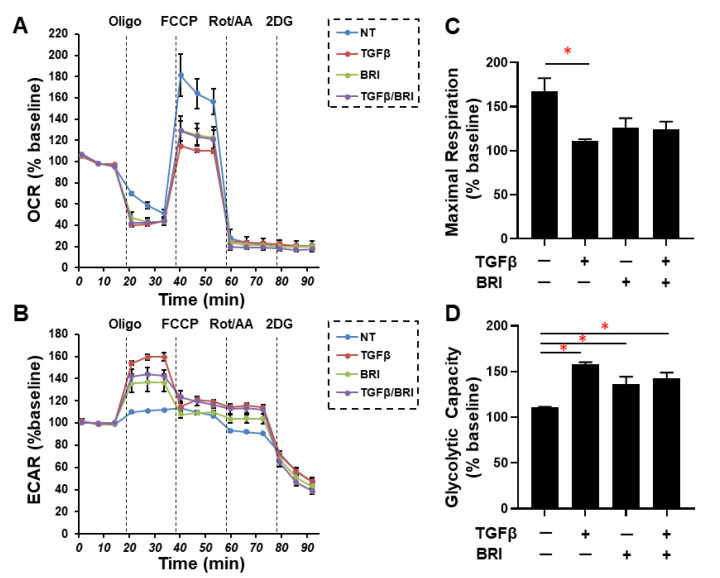
The 2D HTM cells on day 6 untreated (NT) or treated with 5 ng/mL TGF-β2 (TGFβ) in the presence or absence of 10 µM brimonidine (BRI) were subjected to a Seahorse XFe96 Bioanalyzer to evaluate the real-time metabolic cellular functions. The oxygen consumption rate (OCR, panel (**A**)) and extracellular acidification rate (ECAR, panel (**B**)) were simultaneously measured at the baseline, and thereafter, they were further measured by subsequent supplementation with oligomycin (Oligo: complex V inhibitor), FCCP (a protonphore), and rotenone/antimycin A (complex I/III inhibitors) and 2-DG (hexokinase inhibitor). Values are expressed as 100% of the baseline. Maximal respiration (**C**) was defined by OCR with FCCP. Glycolytic capacity (**D**) was defined by ECAR with oligomycin. “+” is reagents addition. “−” is reagents non-addition. All experiments were conducted in triplicates using freshly prepared samples (*n* = 4). Regarding the data, the mean ± standard error of the mean (SEM) are shown. * *p* < 0.05; ANOVA followed by a Tukey’s multiple comparison test.

**Figure 3 bioengineering-09-00310-f003:**
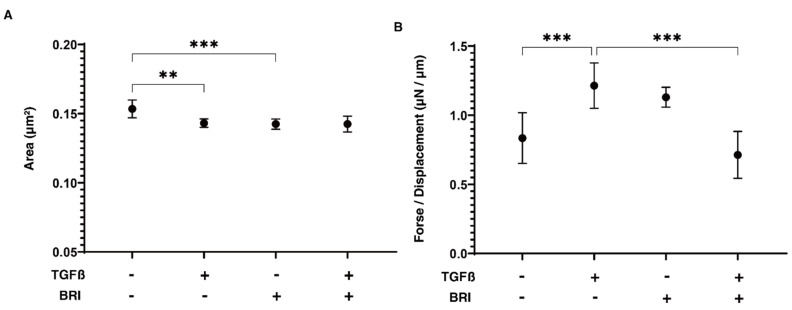
Effects of brimonidine (BRI) on the size (**A**) and hardness (**B**) of the TGF-β2-treated 3D HTM spheroids.Regarding the 3D-cultured HTM for 6 days that were untreated (NT) or treated with 5 ng/mL TGF-β2 (TGFβ) in the absence or presence of 10 μM brimonidine (BRI), the mean sizes of the 3D HTM spheroids are plotted in panel A. Under these conditions, as the stiffness of the 3D HTM spheroid, the required force (μN) to deform the spheroid to the half diameter was measured using a micro-squeezer, and the force/displacement (μN/μm) values are plotted in panel B. “+” is reagents addition. “−” is reagents non-addition. These experiments were performed in triplicate using fresh preparations (*n* = 10 and 15 for size measurement and stiffness analysis, respectively). Experiments were performed in triplicate using fresh preparations (*n* = 4). Regarding the data, the arithmetic mean ± standard error of the mean (SEM) are shown (** *p* < 0.01, *** *p* < 0.005, ANOVA followed by a Tukey’s multiple comparison test).

**Figure 4 bioengineering-09-00310-f004:**
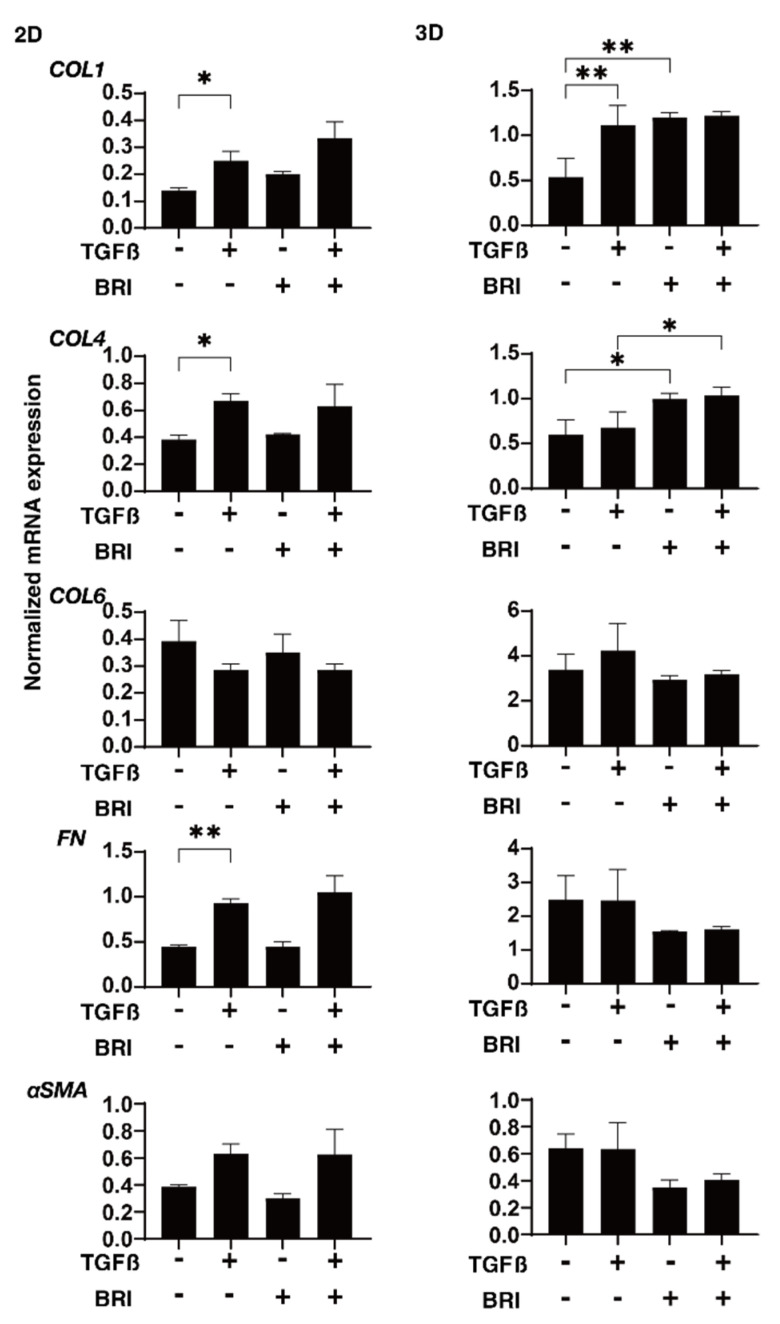
Effects of brimonidine (BRI) on the gene expression of ECM molecules in TGF-β2-treated 2D and 3D HTM cells.The 2D- and 3D-cultured HTM on day 6 were untreated (NT) or treated with 5 ng/mL TGF-β2 (TGFβ) in the presence or absence of 10 μM brimonidine (BRI) and were applied to a qPCR analysis to evaluate the mRNA expression of ECMs, including *COL1 4* and *6*, *FN*, and *a-SMA*. “+” is reagents addition. “−” is reagents non-addition. All analyses were conducted in triplicate using freshly prepared 2D cells (*n* = 5) and 3D spheroids (*n* = 15–20) in each experiment. Regarding the data, the mean ± standard error of the mean (SEM) are shown. * *p* < 0.05, ** *p* < 0.01; ANOVA followed by a Tukey’s multiple comparison test.

**Figure 5 bioengineering-09-00310-f005:**
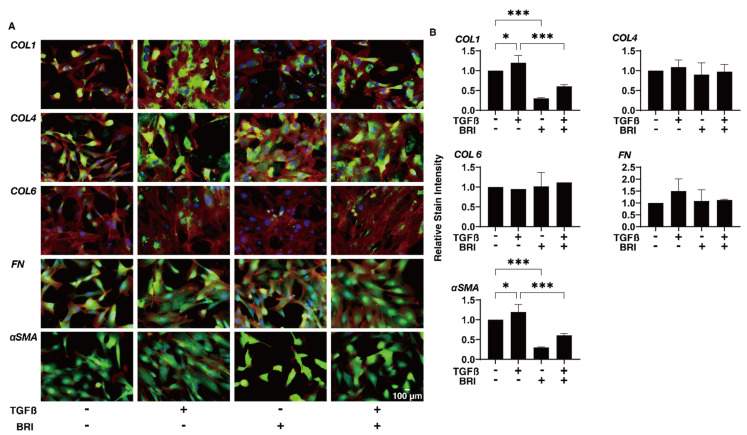
Effects of brimonidine (BRI) on the immunolabeling of ECM molecules in TGF-β2-treated 2D HTM cells.The 2D-cultured HTM on day 6 were untreated (NT) or treated with 5 ng/mL TGF-β2 (TGFβ) in the presence or absence of 10 μM brimonidine (BRI) and were subjected to immunofluorescent labeling of ECMs (*COL1*, *4*, or *6*, *FN*, or *a-SMA*). Experiments were conducted in duplicate (*n* = 5). Representative merged images with DAPI (blue), phalloidin (red), and anti-ECM images (green) are demonstrated (panel A, scale bar; 100 µm). “+” is reagents addition. “−” is reagents non-addition.The staining intensity ratios relative to NT are plotted in panel B. Regarding the data, the mean ± standard error of the mean (SEM) are shown. * *p* < 0.05, *** *p* < 0.005; ANOVA followed by a Tukey’s multiple comparison test.

**Figure 6 bioengineering-09-00310-f006:**
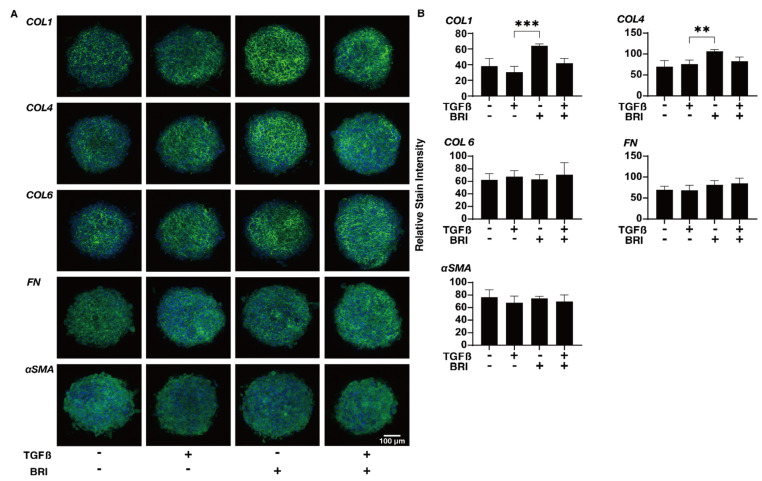
Effects of brimonidine (BRI) on the immunolabeling of ECM molecules in TGF-β2-treated 3D HTM cells.The 3D-cultured HTM on day 6 were untreated (NT) or treated with 5 ng/mL TGF-β2 (TGFβ) in the presence or absence of 10 μM brimonidine (BRI) and were subjected to immunofluorescent labeling of ECMs, including *COL1*, *4*, or *6*; *FN*; or *a-SMA*. Experiments were repeated in duplicate (*n* = 10). Representative merged images with DAPI (blue) and anti-ECM images (green) are demonstrated (panel A, scale bar; 100 μm). The staining intensity ratios relative to NT are plotted (panel B). “+” is reagents addition. “−” is reagents non-addition. Regarding the data, the mean ± standard error of the mean (SEM) are shown. ** *p* < 0.01, *** *p* < 0.005; ANOVA followed by a Tukey’s multiple comparison test.

**Figure 7 bioengineering-09-00310-f007:**
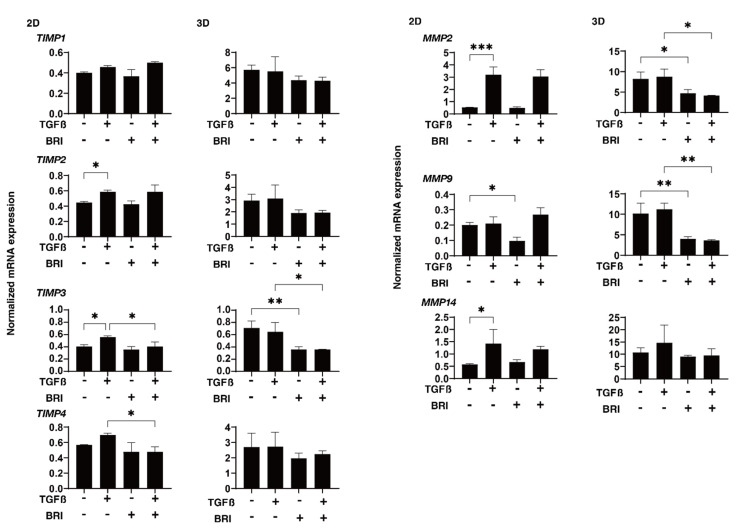
Effects of brimonidine (BRI) on the gene expression of TIMPs and MMPs in TGF-β2-treated 2D and 3D HTM cells. The 2D- and 3D-cultured HTM on day 6 were untreated (NT) or treated with 5 ng/mL TGF-β2 (TGFβ) in the presence or absence of 10 μM brimonidine (BRI) and were applied to a qPCR analysis to evaluate the gene expression of *TIMP1-4* and *MMP 2*, *9*, and *14*. All experiments were conducted in triplicate using freshly prepared samples (2D cells; *n* = 5 and 3D spheroids; *n* = 15–20) in each experimental condition. “+” is reagents addition. “−” is reagents non-addition. Regarding the data, the mean ± standard error of the mean (SEM) are shown. * *p* < 0.05, ** *p* < 0.01, *** *p* < 0.005; ANOVA followed by a Tukey’s multiple comparison test.

**Figure 8 bioengineering-09-00310-f008:**
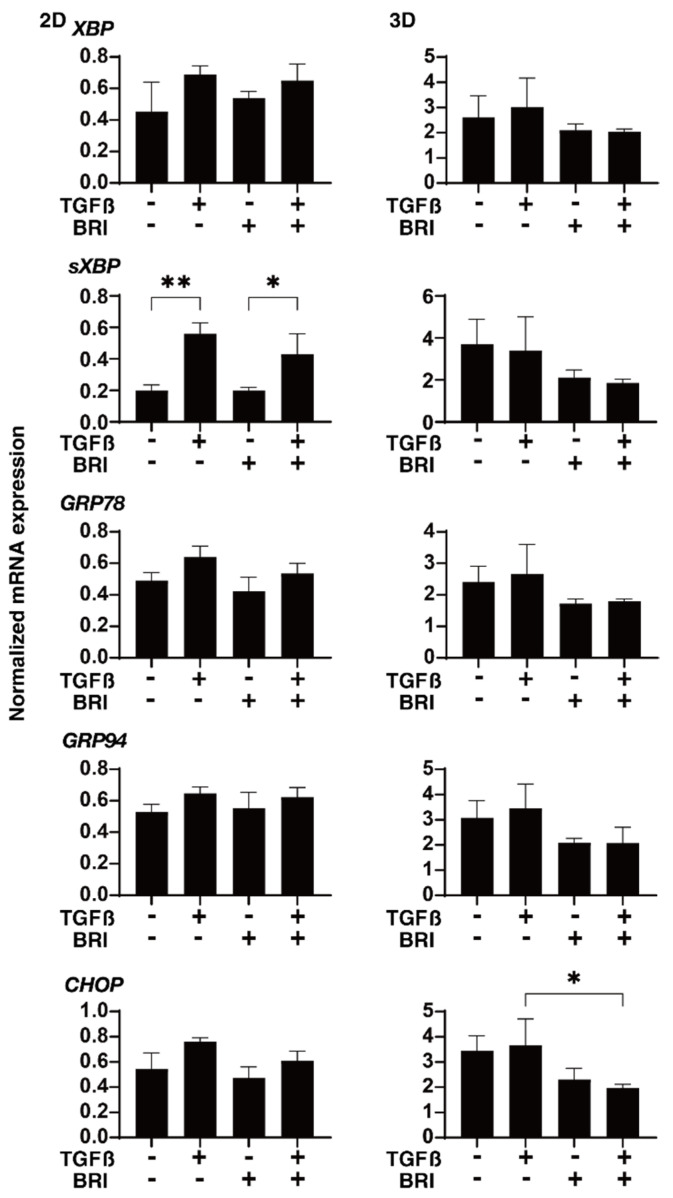
Effects of brimonidine (BRI) on the gene expression of major ER stress-related genes in TGF-β2-treated 2D and 3D HTM cells. The 2D- and 3D-cultured HTM on day 6 were untreated (NT) or treated with 5 ng/mL TGF-β2 (TGFβ) in the presence or absence of 10 μM brimonidine (BRI) and were applied to qPCR analysis to evaluate the gene expression of major ER stress-related genes, including the XBP1, spliced XBP1 (sXBP1) GRP78, GRP94, and CHOP. All experiments were conducted in triplicate using freshly prepared samples (2D cells; *n* = 5 and 3D spheroids; *n* = 15–20) in each experimental condition. “+” is reagents addition. “−” is reagents non-addition. Regarding the data, the mean ± standard error of the mean (SEM) are shown. * *p* < 0.05, ** *p* < 0.01; ANOVA followed by a Tukey’s multiple comparison test.

## Data Availability

The data that support the findings of this study are available from the corresponding author upon reasonable request.

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
