# Peer review of "An α2-Adrenergic Agonist, Brimonidine, Beneficially Affects the TGF-β2-Treated Cellular Properties in an In Vitro Culture Model"

_bioengineering, 2022, doi:10.3390/bioengineering9070310_

Round 1
Reviewer 1 Report
This is a very interesting paper where the authors report the effects of brimonidine (BRI) on 2D and 3D cell cultures using human trabecular meshwork (HTM) cells treated with TGF-beta2.
The descriptions of cell culture and treatments are correct. I wonder the number of experiments performed in the cell cultures treatment experiments, the passage number of the cells and if cell culture medium is FBS supplemented.
Author Response
Dear Editor,
Thank you very much for the constructive comments concerning our manuscript, " An a2-adrenergic agonist, brimonidine, beneficially affects the TGF-β2-induced cellular properties in vitro human trabecular meshwork (HTM) culture model”. We carefully checked all of the Reviewer’s comments and prepared a revised version of our paper that takes these comments into account. The changes are listed below.
Reviewer 1
This is a very interesting paper where the authors report the effects of brimonidine (BRI) on 2D and 3D cell cultures using human trabecular meshwork (HTM) cells treated with TGF-beta2.
- The descriptions of cell culture and treatments are correct. I wonder the number of experiments performed in the cell cultures treatment experiments, the passage number of the cells and if cell culture medium is FBS supplemented.
Answer; Thank you for this comment. As suggested, requested information is included in the Method; “The immortalized cells of the HTM (Applied Biological Materials Inc., Richmond Canada) were maintained in the culture medium composed of HG-DMEM containing 10 % FBS, 1 % L-glutamine, 1 % antibiotic-antimycotic up to passage 20 as according the conventional 2D and 3D spheroid culture conditions during 6 days as followed in a previous report11, after checking these cells were truly HTM cells 12. All 2D and 3D cell cultures were repeated three times to subject following analyses. For the evaluation of the drug efficacy of brimonidine (BRI), an a2-adrenergic agonist, on TGF-β2 (5 ng/mL) treated HTM cells (2D and 3D), 10 mM BRI were supplemented at cell culture Day 1. The drug concentrations were confirmed as the optimal concentrations based on previous studies 11,13,14, and a cell viability analysis of BRI using retinal pigment epithelium cells 15.”.
Reviewer 2
The present study examines the effect of brimonidine (BRI) on barrier function and cellular metabolic function in 2D and 3D cultures derived from human trabecular meshwork (HTM).
- The authors claimed that TGFb2-treated HTM spheroids is an established in vitro model of glaucomatous TM (Lines 58-62). However, the characteristics of the HTM spheroids in the current study are different from the authors’ previous publication (citation 11). There are no increases in the expression of collagens 1,4, 6 and fibronectin in the presence of TGFb2, in contrast to the increases observed in citation 11. One would speculate the validity of the TGFb2-treated spheroids as an in vitro model of glaucomatous TM. As such, the authors should clarify such differences between the current and previous findings (citation 11). However, there is little discussion regarding this aspect.
Answer; Thank you for this comment. As suggested, this issue is included in the study limitation in the Discussion; “As study limitation for the current investigation, TGFb2-treated HTM spheroids as an established in vitro model of glaucomatous TM represent quite reproducible experimental observations in terms of their physical properties, size and stiffness as above. However, the characteristics of the HTM spheroids in the current study were some different from our previous publication in terms of the gene expressions of ECMs11. That is, there are no significantly increases in the expression of COL1, 4, 6 and FN in the presence of TGFb2, in contrast to the increases observed in our previous study11. Although we do not exactly know these differences, we speculated that some cell culture conditions such as FBS and other culture medium components even though there were the same numbers of products, in addition to the numbers of cell passages. However, even if there were such diversity between preparations, we believe current data of the BRI induced effects as above because those were performed using the same preparations and repeated at least three times. Therefore, in conclusion, currently observed new beneficial pharmacological effects of BRI toward glaucomatous HTM may provide significant insight into the therapeutic strategy for antiglaucoma medication in patients with POAG. Furthermore, additional studies will be required for a comprehensive understanding of the molecular mechanisms for the BRI induced effects on the dexamethasone treated steroid induced glaucoma TM model, in addition to the TGF-b2 treated POAG TM model to be acquired with certainty.”.
Issues to be addressed by the authors are as follows:
- Introduction: Line 58, It seems that the 3D culture model is referred as an in vivo model. This is incorrect and needs to be rectified because 3D culture is an in vitro model.
Answer; Thank you for this comment. As pointed out, rather than in vivo, “in vitro” should be correct. Therefore, corresponding phrase was changed.
- Method: Two studies were cited for the immunolabeling method, but brief experimental details should be provided about the preparation of 2D and 3D culture, such as fixation, permeabilization, incubation period of primary and secondary antibodies?
Answer; Thank you for this comment. As suggested, more detail method related the immunocytochemistry was included; “Immunocytochemistry of the HTM cells
The immunocytochemistry was performed as described in a recent report 18,19. Briefly, the 2D and 3D cultured HTM cells as above were each fixed with 4 % paraformaldehyde, permeabilized in 0.1 % NP-40 in PBS for 10 min, and incubated successively with the 1st antibody; an anti-human rabbit antibody (1:200 dilutions) against COL1, 4 or 6 (ROCKLAND antibodies & assays, Limerick, PA. U.S.A.) or FN (Santa Cruz Biotechnology, Inc., Dallas, Tx. U.S.A.) for overnight, and a mixture of the 2nd antibody; a goat anti-rabbit IgG (488 nm, 1:1000 dilutions, Invitrogen, Waltham, MA. U.S.A.), phalloidin (594 nm, 1:1000 dilutions, Cayman Chemical, Ann Arbor, MI. U.S.A.) and DAPI (1:1000 dilutions, DOJINDO, Osaka, Japan) for 3 hours. Thereafter, the confocal immunofluorescent images were obtained.”.
- Results: Line 140: Revise this phrase. Why the term 2) is used here?
Answer; Thank you for this comment. As pointed out, this is careless mistake, and therefore, this sentence was divided into two sentences; “The 3D spheroids were reduced in their sizes and became harder as the result of the 5 ng/ml TGF-b2 or 10 mM BRI. In addition, BRI induced a significant reduction in the elevated TGF-b2 induced hardness of the 3D spheroids without synergist effects toward the sizes of the 3D spheroids.”.
- Figures 1, 3, 4, 5, 6,7,8: make the labelling uniform across the figures. Place negative to indicate no treatment. For example, put – for TGF and – for BRI in the first column for untreated cells. See Figure 2.
Answer; Thank you for this comment. As suggested, the labelling of the figures were revised in a uniformed fashion as suggested.
- There is no description of Figure 3 results.
Answer; Thank you for this comment. As pointed out, description of Fig. 3 was included within the results; “Within the 3D HTM spheroid models, similar but also some different physical effects by BRI toward TGF-b2 induced effects were also observed (Fig. 3), which were thought to be a representative in vitro model that replicates multi-sheet layers of HTM structure 11,20. That is, the 3D spheroids were reduced in their sizes and became harder as the result of the 5 ng/ml TGF-b2 or 10 mM BRI. In addition, BRI induced a significant reduction in the elevated TGF-b2 induced hardness of the 3D spheroids without synergist effects toward the sizes of the 3D spheroids (Fig. 3).”
- Figure 8: Indicate which graphs are data derived from 2D and 3D. Line 191-196: The legend should be concise to include essential details.
Answer; Thank you for this comment. As pointed out, information related to 2D and 3D was included within this figure.
- Figure 5B: Revise the title. Is this the intensity of immunolabelling or normalisation of mRNA?
Answer; Thank you for this comment. As pointed out, title of panel B was changed to “relative staining intensities”.
- Discussion: There is no discussion about the lack of effects of BRI on the synthesis of ECM as demonstrated by Figure 3, 4, 5. Zhang et al. (Sci Rep. 2020; 10: 20292) recently compared the effect of TGFb on expression of ECM and a-SMA between 2D and 3D HTM cell cultures. The authors should discuss the differences between the current study and Zhang et al.
Answer; Thank you for this comment. As suggested, discussion related to Sci Rep. 2020; 10: 20292 is included within the study limitation in the Discussion; “As study limitation for the current investigation, TGFb2-treated HTM spheroids as an established in vitro model of glaucomatous TM represent quite reproducible experimental observations in terms of their physical properties, size and stiffness as above. However, the characteristics of the HTM spheroids in the current study were some different from our previous publication in terms of the gene expressions of ECMs11. That is, there are no significantly increases in the expression of COL1, 4, 6 and FN in the presence of TGFb2, in contrast to the increases observed in our previous study11. Although we do not exactly know these differences, we speculated that some cell culture conditions such as FBS and other culture medium components even though there were the same numbers of products, in addition to the numbers of cell passages. However, even if there were such diversity between preparations, we believe current data of the BRI induced effects as above because those were performed using the same preparations and repeated at least three times. Therefore, in conclusion, currently observed new beneficial pharmacological effects of BRI toward glaucomatous HTM may provide significant insight into the therapeutic strategy for antiglaucoma medication in patients with POAG. Furthermore, additional studies will be required for a comprehensive understanding of the molecular mechanisms for the BRI induced effects on the dexamethasone treated steroid induced glaucoma TM model, in addition to the TGF-b2 treated POAG TM model to be acquired with certainty.”
Reviewer 3
The authors presented an interesting study evaluating the effects of brimonidine on 2D and 3D cell cultures using human trabecular meshwork treated with TGF-β2. The manuscript is with merit and the findings are worth reporting, but the authors should address the following comments before publication.
- Title; The authors should revise the title and eliminate the abbreviation “HTM”
Answer; Thank you for this comment. As pointed out, the abbreviation “HTM” was eliminated.
- Abstract; Revise the entire manuscript the use of abbreviations: an abbreviation should be explained once in the manuscript the first time that it is used – revise the manuscript correspondingly (i.e. line 17 the explanations of the abbreviation “2D and 3D” should be provided)
Answer; Thank you for this comment. As pointed out, the abbreviation of 2D and 3D at line 17 was included and also other abbreviations were carefully checked.
- Introduction; Revise the entire manuscript the use of abbreviations: an abbreviation should be explained once in the manuscript and after only the abbreviation (and not the full explanation) should be used – revise the manuscript correspondingly (i.e. 45 and 48 the explanations of the abbreviation “RGC” should not repeated twice in the following line, only the abbreviation RGC should be used after)
Answer; Thank you for this comment. As pointed out, the abbreviation of RGS at line 48 was eliminated was included and also other abbreviations were carefully checked.
- Discussion; The authors should add a “limitations” section before the conclusions of the manuscript.
- In the conclusive paragraph, the authors should provide some insight about the clinical applicability of their findings in the clinical practice and about the future directions of their research.
Answer for 4 and 5; Thank you for this comment. As suggested, study limitation for the present study is included in the Discussion; “As study limitation for the current investigation, TGFb2-treated HTM spheroids as an established in vitro model of glaucomatous TM represent quite reproducible experimental observations in terms of their physical properties, size and stiffness as above. However, the characteristics of the HTM spheroids in the current study were some different from our previous publication in terms of the gene expressions of ECMs11. That is, there are no significantly increases in the expression of COL1, 4, 6 and FN in the presence of TGFb2, in contrast to the increases observed in our previous study11. Although we do not exactly know these differences, we speculated that some cell culture conditions such as FBS and other culture medium components even though there were the same numbers of products, in addition to the numbers of cell passages. However, even if there were such diversity between preparations, we believe current data of the BRI induced effects as above because those were performed using the same preparations and repeated at least three times. Therefore, in conclusion, currently observed new beneficial pharmacological effects of BRI toward glaucomatous HTM may provide significant insight into the therapeutic strategy for antiglaucoma medication in patients with POAG. Furthermore, additional studies will be required for a comprehensive understanding of the molecular mechanisms for the BRI induced effects on the dexamethasone treated steroid induced glaucoma TM model, in addition to the TGF-b2 treated POAG TM model to be acquired with certainty.”.

Reviewer 2 Report
The present study examines the effect of brimonidine (BRI) on barrier function and cellular metabolic function in 2D and 3D cultures derived from human trabecular meshwork (HTM). The authors claimed that TGFb2-treated HTM spheroids is an established in vitro model of glaucomatous TM (Lines 58-62). However, the characteristics of the HTM spheroids in the current study are different from the authors’ previous publication (citation 11). There are no increases in the expression of collagens 1,4, 6 and fibronectin in the presence of TGFb2, in contrast to the increases observed in citation 11. One would speculate the validity of the TGFb2-treated spheroids as an in vitro model of glaucomatous TM. As such, the authors should clarify such differences between the current and previous findings (citation 11). However, there is little discussion regarding this aspect.
Issues to be addressed by the authors are as follows:
Introduction:
Line 58, It seems that the 3D culture model is referred as an in vivo model. This is incorrect and needs to be rectified because 3D culture is an in vitro model.
Method:
Two studies were cited for the immunolabeling method, but brief experimental details should be provided about the preparation of 2D and 3D culture, such as fixation, permeabilization, incubation period of primary and secondary antibodies?
Results:
Line 140: Revise this phrase. Why the term 2) is used here?
Figures 1, 3, 4, 5, 6,7,8: make the labelling uniform across the figures. Place negative to indicate no treatment. For example, put – for TGF and – for BRI in the first column for untreated cells. See Figure 2.
There is no description of Figure 3 results.
Figure 8: Indicate which graphs are data derived from 2D and 3D.
Line 191-196: The legend should be concise to include essential details.
Figure 5B: Revise the title. Is this the intensity of immunolabelling or normalisation of mRNA?
Discussion:
There is no discussion about the lack of effects of BRI on the synthesis of ECM as demonstrated by Figure 3, 4, 5.
Zhang et al. (Sci Rep. 2020; 10: 20292) recently compared the effect of TGFb on expression of ECM and a-SMA between 2D and 3D HTM cell cultures. The authors should discuss the differences between the current study and Zhang et al.
Author Response

(The authors gave the same response as above.)

Reviewer 3 Report
The authors presented an interesting study evaluating the effects of brimonidine on 2D and 3D cell cultures using human trabecular meshwork treated with TGF-β2. The manuscript is with merit and the findings are worth reporting, but the authors should address the following comments before publication.
Title
- The authors should revise the title and eliminate the abbreviation “HTM”
Abstract
- Revise the entire manuscript the use of abbreviations: an abbreviation should be explained once in the manuscript the first time that it is used – revise the manuscript correspondingly (i.e. line 17 the explanations of the abbreviation “2D and 3D” should be provided)
Introduction
- Revise the entire manuscript the use of abbreviations: an abbreviation should be explained once in the manuscript and after only the abbreviation (and not the full explanation) should be used – revise the manuscript correspondingly (i.e. 45 and 48 the explanations of the abbreviation “RGC” should not repeated twice in the following line, only the abbreviation RGC should be used after)
Discussion
- The authors should add a “limitations” section before the conclusions of the manuscript
- In the conclusive paragraph, the authors should provide some insight about the clinical applicability of their findings in the clinical practice and about the future directions of their research
Author Response

(The authors gave the same response as above.)

Round 2
Reviewer 2 Report
The authors have addressed the comments.
Reviewer 3 Report
The authors addressed al my comments and the manuscript in my opinion can be accepted for publication.